# Subaqueous Topographic Deformation in Abandoned Delta Lobes—A Case Study in the Yellow River Delta, China

**Yunfeng Zhang [1,2], Yingying Chai [3,4], Caiping Hu [1,2], Yijun Xu [3,\*], Yuyan Zhou [5], Huanliang Chen [1,2], Zijun Li [4], Shenting Gang [1,2] and Shuwei Zheng [1,4,\*]**

1   801 Institute of Hydrogeology and Engineering Geology, Shandong Provincial Bureau of Geology & Mineral Resources, Jinan 250014, China
2   Shandong Engineering Research Center for Environmental Protection and Remediation on Groundwater, Jinan 250014, China
3   Coastal Studies Institute, Louisiana State University, Baton Rouge, LA 70803, USA
4   College of Geography and Environment, Shandong Normal University, Jinan 250358, China
5   State Key Laboratory of Simulation and Regulation of Water Cycle in River Basin, China Institute of Water Resources and Hydropower Research, Beijing 100048, China
\*   Correspondence: yjxu@lsu.edu (Y.X.); 618067@sdnu.edu.cn (S.Z.)

**Abstract:** Reduction in river discharge and sediment load has left deltaic lobes in the world's many river deltas starving, but knowledge of how the subaqueous topography of these abandoned subdeltas responds to environmental changes is limited. In this study, we aimed to determine the long-term dynamics of the subaqueous seabed of abandoned delta lobes to advance current knowledge. As a case study, we selected an abandoned subdelta on the Yellow River Delta of the Bohai Sea, China, for which three-decade long (1984–2017) bathymetric data and long-term river discharge and sediment load records are available. We analyzed the seafloor surface change and quantified the void space from the sea water surface to the seafloor. In addition, we surveyed the seafloor surface with an M80 unmanned surface vehicle carrying a multibeam echo sounder system (MBES) in 2019 to obtain high-resolution microtopography information. We found that a net volume of $5.3 \times 10^8$ m$^3$ of sediment was eroded from the study seabed within an area of $3.6 \times 10^8$ m$^2$ during 1984–2017. This volumetric quantity is equivalent to 6.89 billion metric tons of sediment, assuming a bulk density of 1.3 t/m$^3$ for the seabed sediment. The seabed erosion from 0 to $-5$ m, from $-5$ to $-10$ m, and below $-10$ m has showed a similar increasing trend over the past 33 years. These findings suggest that seabed erosion in this abandoned subdelta will very likely continue, and that other abandoned delta lobes in the world may have been experiencing similar seabed erosion due to the interruption of the sediment supply and sea level rise. It is not clear if the seabed erosion of abandoned delta lobes would have any effect on the stability of the coastal shoreline and continental shelf.

**Keywords:** seabed erosion; sediment budget; anthropogenic disturbance; multibeam echo sounder; Yellow River Delta



## 1. Introduction

River and coastal systems are some of the most economically vital [1] and socially important areas on Earth [2]. For example, 44% of the world's population resides within 150 km of the coastline [3], and river deltas house over 5% of the global population but account for less than 0.5% of the world's land area [4]. Unfortunately, as some of the most densely populated areas, the natural environments of many river deltas have changed due to both natural and anthropogenic factors [5,6], such as tropical storms [7,8], river engineering [9–11], floods [12,13], and sea level rise [14,15]. Therefore, knowledge of the evolution processes of delta geomorphology is critical for understanding of how river-coastal systems respond to natural and anthropogenic effects [16,17].

Global deltas have achieved an annual land gain of $54 \pm 12$ km$^2$ over the past 30 years [18]. However, the declining sediment load, estuarine engineering, and climate change have altered riverine sediment supply and distribution in many of the world's large rivers [19,20], leading to their deltaic land loss [18]. Nienhuis (2020) pointed out that nearly 1000 deltas collectively lost $12 \pm 3.5$ km$^2$ of land due to a decrease in the sediment supply caused by river damming [18]. In particular, there are several large deltas that have been strongly affected by river damming, such as the Mekong [5], Yangtze [4], Mississippi [20], Nile [21], and Red [6] River Deltas. Large-scale projects in estuary and delta areas, including land reclamation and reservoir construction, have also impacted delta morphodynamics [4], with maximum local erosion depths greater than 10 m in the North Channel of the Yangtze River Delta [9]. As riverine sediment loads decline, previous delta lobes and the bifurcated channels become abandoned. Few studies have investigated how these abandoned deltaic lands develop as global sea levels continue rising and the frequency and intensity of storms increase. Such information can be crucial for developing effective management strategies and plans for river deltas.

This study was, therefore, conducted to gain knowledge about the long-term geomorphic dynamics of abandoned river deltas and their bifurcation channels. As a case study, we analyzed 34 years of historical bathymetric data as well as recently-collected sediment samples from several abandoned subdeltas on the Yellow River Delta. The main objectives of our study are to (1) obtain the high-resolution seabed microtopography characteristics of an abandoned delta, (2) examine the evolution process of the abandoned delta corresponding to a lack of sediment supply over the past three decades, and (3) discuss the evolution trend of the seabed in the abandoned subaqueous delta. Findings gained from this study will gain insight into the development of the abandoned subaqueous Yellow River Delta. Such information can be useful for developing effective management strategies and plans for the Yellow River Delta, as well as for other alluvial river deltas in the world.

## 2. Study Area

The Yellow River (Huanghe) Delta is a typical river-dominated delta located at the western Bohai Sea, China [22]. Its modern delta was formed in 1855 when the river changed its course from the Yellow Sea to the Bohai Sea [23,24]. Currently, the Yellow River Delta is rich in wetlands, oilfields, and aquaculture [25]. Similar to other mega-deltas, the Yellow River Delta is significantly influenced by climate forcing, in particular sea level rise, as well as river engineering practices including river diversion, channelization, and construction of dams, dikes, and levees [26]. However, its deltaic development is distinctively different from other deltas as the river has frequently changed its course in the past [26]. Since 1855, the Yellow River has avulsed more than eleven times, far more frequently than any other large rivers in the world have. Therefore, some of the subaerial deltas associated with its current course are rapidly expanding toward the sea and the abandoned subaqueous deltas are retreating [27]. For example, most of the abandoned Yellow River delta lobes have turned to slight erosion after 2000 [1]. On the whole, the subaqueous Yellow River Delta experienced a change from deposition to erosion after 2005 [22]. The upstream sediment load of $0.48 \times 10^8$ t/a may maintain the equilibrium of the Yellow River Delta [28]. The above studies have analyzed the geomorphic processes of the abandoned Yellow River Delta and have deepened the understanding of the evolutionary mechanism of the abandoned Yellow River Delta. However, high-resolution data on the microtopography and seabed evolution process of these abandoned subaqueous delta lobes are lacking, and this information is significant for forecasting the evolution trend and equilibrium state of the abandoned delta lobes, as well as for the whole Yellow River Delta.

The Yellow River Delta is located in the western Bohai Sea (Figure 1a) [29]. Four artificial flood control measures were implemented in 1953, 1964, 1976, and 1996, which formed the Shenxiangou (SXG, 1953–1964), Diaokouhe (DKH, 1964–1976), Qingshuigou (QSG, 1976–1996), and Q8 channels (consisting of the abandoned course, running from 1996–2007 and the present course, running from 2007 to the present), respectively (Figure 1b). It

belongs to the warm temperate monsoon climate zone, with an annual mean rainfall of 590.9 mm [16]. The southerly winds prevail with a range of 3–6 m/s in summer and northerly winds prevail with >10 m/s in winter [30]. The mean wave height is up to 1.5 m [30]. The mean tidal range near the shoreline is 0.6–0.8 m in the YRD [16]. It was once famous for its high suspended sediment concentration (26.5 g/L) [24]. However, the sediment load has experienced a stepwise reduction due to dam construction since the 1950s [31,32]. In this study, we selected the section between the SXG and DKH subdelta areas with an area of 360 km² (Figure 1c) as a key area to analyze the seabed deformation process and causes after the Yellow River course was abandoned in 1976. The mean tidal range is 1.4 m [33], and the mean significant wave height is 1.5 m [34].

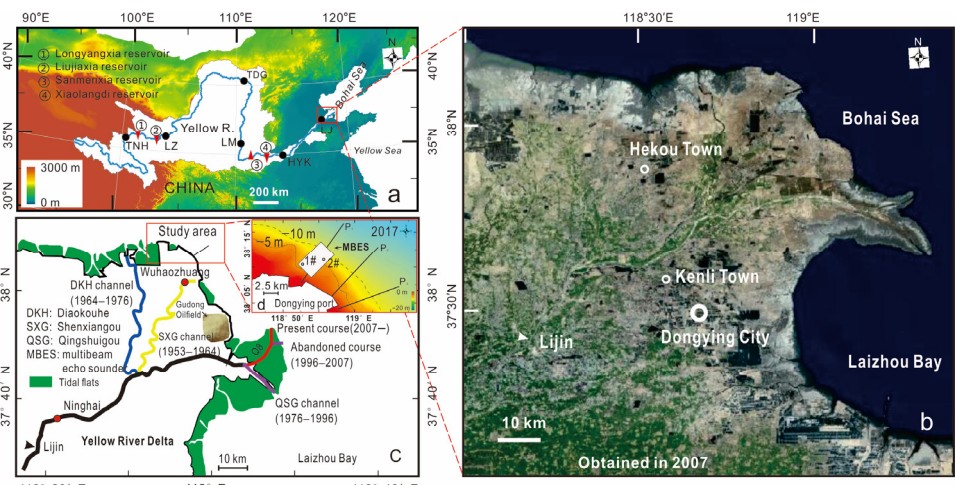

**Figure 1.** (**a**) Sketch maps of the Yellow River Basin. Black dots represent the hydrological stations in the upper Yellow River, including Tangnaihai (TNH), Lanzou (LZ), and Toudaoguai (TDG); the middle Yellow River, including Longmen (LM); and the lower Yellow River, including Huayuankou (HYK) and Lijin (LJ). Red triangles represent large engineering projects, including dams and reservoirs. (**b**) Map of the modern Yellow River Delta. Satellite image obtained from https://map.tianditu.gov.cn/ (accessed on 29 March 2023). Drawing Review Number is GS (2021) 3715. Ministry of Natural Resources of the People's Republic of China. (**c**) Map of the modern Yellow River Delta. The colored lines show the most recent channel shifts. (**d**) Bathymetry of Dongying Port and the nearby seabed in 2017. The white rectangle with a black border shows the MBES (multibeam echo sounder) survey area in September 2019. Three black lines labeled $P_1$, $P_2$, and $P_3$ show the cross-section profiles of the seabed. Black hollow circles show the seabed sampling locations. 1# and 2# show the locations of sediment samples.

## 3. Materials and Methods

### 3.1. Historical Bathymetric Data and Current Microtopography Survey

In this study, we collected bathymetric charts surveyed by the Navigation Guarantee Department of the Chinese Navy Headquarters (surveyed in 1984 and 2007 at a scale of 1:250,000) and the Maritime Safety Administration of the People's Republic of China (surveyed in 2017 at a scale of 1:25,000). These surveys were performed with dual-frequency echo sounders and GPS positioning. The vertical error of water depth (H) is 0.2 m and $\pm 0.1$ H, when H $\leq$ 10 m and H > 20 m, respectively [9]. Data collection quality referenced the Codes for Survey of Port and Waterway Engineering (http://zs.mot.gov.cn, (accessed on 15 May 2019)). The reference datum of these charts is the 1985 National Elevation Datum, China.

In September 2019, we used an M80 unmanned surface vehicle (USV, Yunzhou, China) with a Seabat T50 multibeam echo sounder system (MBES) to gain high-resolution bathymetric data from the study area (Figure 1c). A rectangular area 500 m wide and 800 m long was chosen for the survey to analyze the characteristics of the seabed surface microtopography.

The M80 USV had a length of 5.6 m and a width of 2.4 m. The speed of the USV was controlled to be as constant as possible at 2–3 m s$^{-1}$ during MBES data collection The frequency of the Seabat T50 MBES was 400 kHz during the survey with the vertical error at a depth of 6 mm. The equal-distance model provides a wide survey range of 150° in the Teledyne PDS control center, and the swath width covered approximately 6 times the instantaneous water depth. A Trimble real-time differential global positioning system (DGPS) was used to control the above data position error at the decimeter level [35].

### 3.2. Seabed Sediment Sampling and Analysis

Sediment samples were collected from the seabed surface at two locations in 2019 (Figure 1c). Sampling location 1 is located near the coastal line, and sampling location 2 was far from the coastal shoreline. During sampling, the surveying vessel was stopped, and a grab sampler which collected the top 3–10 cm of the seabed sediment was used. An amount of 10% $H_2O_2$ and 10% HCl was used to remove organic matter and $CaCO_3$ concretions. An appropriate amount of $(Na_3PO_3)_6$ was used to disperse the sediment samples. Then, the sediment samples were analyzed in a laboratory with a Mastersizer 2000 laser granulometer (Malvern Panalytical Ltd., Malvern, UK).

### 3.3. Quantification of Seabed Deformation

In this study, the bathymetric charts were georeferenced using 6–9 fixed landmarks in ArcGIS 10.3 [4]. Subsequently, the water depth points were collected from bathymetric charts. These points included original water depth points (adjacent cross-sections were spaced 1 km apart, having 250–450 m intervals along each cross-section) and water depth points on the isobaths of 0 m, −2 m, −5 m, −10 m, and −20 m (each point-interval along isobaths were approximately 300–500 m). As a result, each digitized bathymetric chart has a density of 70–100 points per square kilometer. The ordinary kriging interpolation method was used to generate a digital elevation model (DEM) with a resolution of 100 × 100 m in ArcGIS 10.3 (ESRI, Redlands, CA, USA). The Kriging method has been widely used and is highly effective in the analysis of volume from the water surface to the seafloor [35–37]. The interpolation error is mainly from bathymetric chart data and the interpolation method (kriging interpolation method). Overall, the error of the DEMs was estimated to be less than 10% [9]. In addition, the mean rate of sea level rise (this value includes sediment compaction and land subsidence) is 3.6 mm/a during 1980–2021 in the study area (https://www.mnr.gov.cn, (accessed on 22 May 2022)). In this study, we selected this value as the seabed volume change caused by sea level rise and land subsidence. The study aera is 3.6 × 10$^8$ m$^2$. Thus, the seabed volume change caused by sea level rise and land subsidence ($V_s$) is 0.31 × 10$^8$ m$^3$ and 0.14 × 10$^8$ m$^3$ during 1984–2007 and 2007–2017, respectively. The basic assumption is that the void volume from the water surface to the seafloor changes primarily due to changes in depth caused by erosion or deposition of materials on the seabed and relative sea level rise (including land subsidence) [38–40]. Thus, the changes in the depth of the void volume from the water surface to the seafloor (water depth) indicate seabed deformation. The reference datum is the 1985 National Elevation Datum of China (0 m was calibrated to the mean sea level). Furthermore, sediment mass (*M*) can be calculated according to the change in void volume from the water surface to the seafloor (water volume) (*V*) between adjacent years in the study area (Figure 1c). The water volume changes ($V_c$) were calculated by water volume below 0 m between adjacent years as follows:

$$V_c = V_a - V_b - V_S \tag{1}$$

where $V_a$ and $V_b$ are the water volumes in a certain year, and $V_s$ is the subsidence rate. Sediment mass (*M*) was estimated as follows:

$$M = \rho \times V_a - V_b - V_S/(a-b) \tag{2}$$

where $\rho$ is the dry bulk density of the seabed material. In this study, the dry bulk density was chosen to be 1.3 t/m$^3$ [41]. $a$ and $b$ are the years of bathymetric data collection, and $a$ is one year later than $b$.

The change in mean water depth ($D$) and thickness of seabed deformation ($T$) were calculated by the following formulas:

$$D = V/S \tag{3}$$

where $S$ and $V$ are the seabed area (m$^2$) and water volume (m$^3$) below 0 m in a certain year, respectively, and

$$T = D_a - D_b/(a - b) \tag{4}$$

where $T$ is the thickness of the changed seabed material (unit is m/a). $Da$ and $Db$ are the mean water depths (m) in different years.

Eight MBES survey lines perpendicular to the isobath direction were conducted near the DKH subdelta lobe on 28 September 2019. The water depth data collected by MBES were processed in the PDS 2000 software (PDS V4.1.7.3), including beam calibrations for roll, pitch, and yaw, as well as error beam remove. The tidal correction was performed using Dongying Port data (Figure 1c), and the base level was calibrated to the mean sea level. The final data were exported to ArcGIS 10.3 and formed a grid model with a resolution of 1 m × 1 m.

The slope of MBES data was calculated using the 3D Analyst Tools (Raster Surface-Slope) in ArcGIS 10.3. In this study, each cell was 1 m × 1 m. The slope was calculated by the following formula:

$$Degree = ATAN(h/l) \tag{5}$$

where *Degree* is the slope in degrees (°), $h$ is the height difference between two cells, and $l$ is distance between the two cells.

## 4. Results

### 4.1. Seabed Deformation

The void volume from the water surface to the seabed of the abandoned subdeltas increased by 16% from 1984 to 2017. Specifically, the volume increased from $34.6 \pm 3 \times 10^8$ m$^3$ in 1984 to $39.9 \pm 4 \times 10^8$ m$^3$ in 2017 (Figures 2 and 3a), showing a net erosion of $5.3 \times 10^8$ m$^3$ of seabed material. This volumetric quantity is equivalent to 6.89 billion metric tons of sediment, assuming a bulk density of 1.3 t/m$^3$ for the seabed material (Figure 2). The volumes from 0 to $-5$ m, from $-5$ to $-10$ m, and below $-10$ m showed the same increasing trend (Figure 3a).

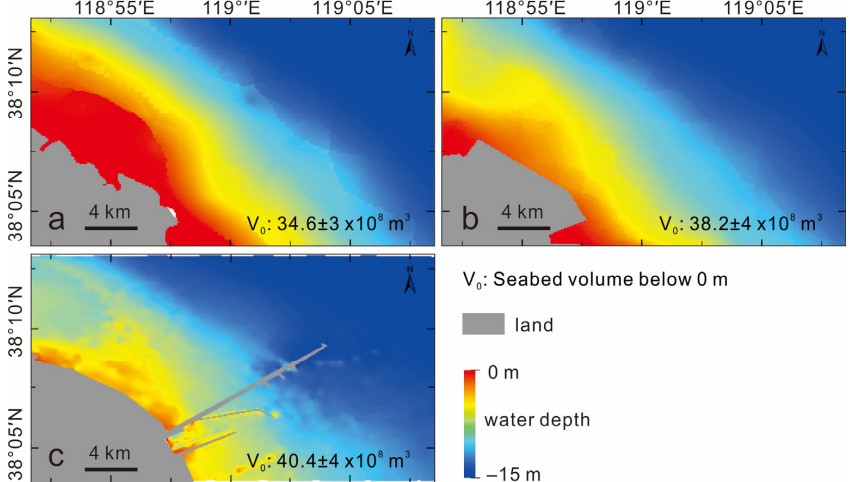

**Figure 2.** The digital elevation model (DEM) of the study area in the Yellow River Delta. (**a–c**) the DEM in 1984, 2007, and 2017.

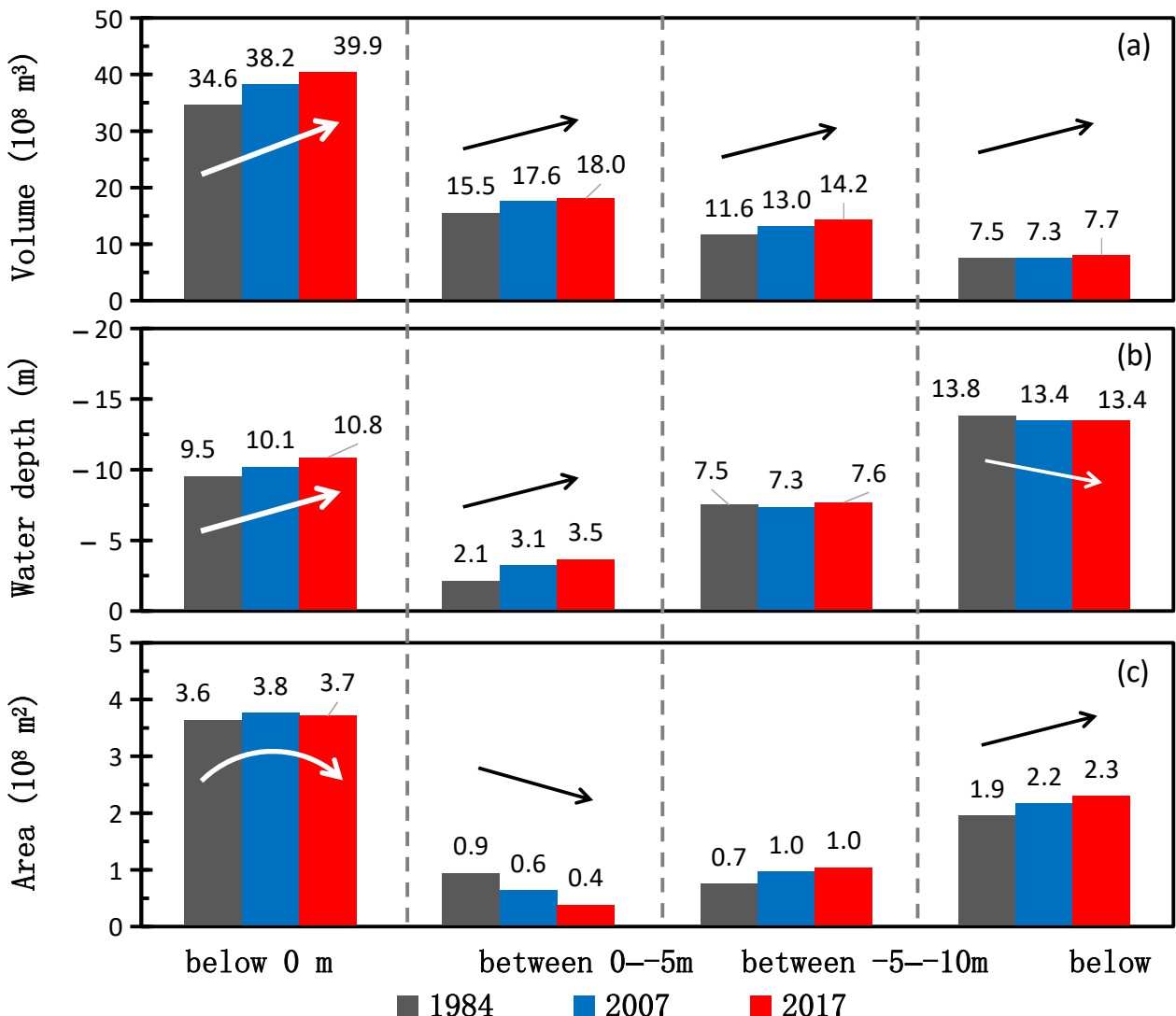

**Figure 3.** The void volume from water surface to the seafloor (**a**), water depth (**b**), and area change (**c**) in the abandoned lobe (area shown in Figure 1c). The white and black arrows indicate changing trends.

Therefore, the mean water depth showed a deepening trend (Figure 3b) with varying deformation intensities. The area from 0 to −5 m showed a constant deepening trend, while the area from −5 to −10 m became shallower from −7.5 m to −7.3 m during 1984 to 2007 and then deepened to −7.6 m in 2017. The area below −10 m did not change after 2007.

The 2D seabed surface area (shown in Figure 1c) first increased from $3.6 \times 10^8$ m$^2$ to $3.8 \times 10^8$ m$^2$ during 1984–2007 and then decreased to $3.7 \times 10^8$ m$^2$ in 2017 (Figure 3c). Specifically, the area from 0 to −5 m shows a decreasing trend from 1984 to 2017, differently from the areas from −5 to −10 m and below −10 m that are characterized by an increasing trend.

Three cross-section profiles (P$_1$–P$_3$, locations in Figure 1c) showed that the −5 m and −10 m isobaths seem to prograde toward the coastline (Figure 4). The −5 m and −10 m isobaths moved approximately 3500 m and 600 m during 2007–2017, respectively (Figure 4a). A similar situation was found in other cross-section profiles (Figure 4b,c).

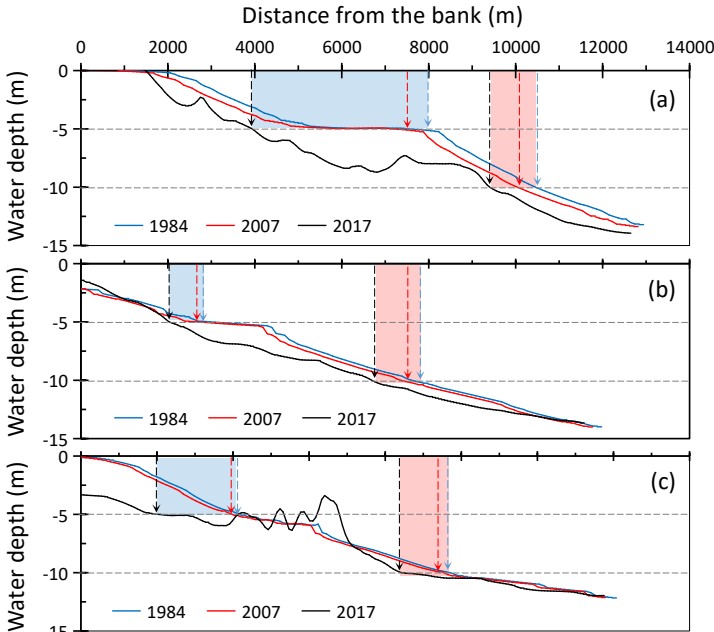

**Figure 4.** Cross-section profile changes in the subaqueous seabed of the SXG-DKH lobes during 1984–2017. (**a**–**c**) correspond to P1, P2, and P3, respectively, in Figure 1c. The green and red blocks show the change in the −5 m and −10 m isobaths from 1984 to 2017, respectively. The blue black, red black, and color lines with arrows (green, red, and black dashed lines indicate the above isobath's location in 1984, 2007, and 2017, respectively) show the locations of the −5 m and −10 m isobaths in 1984, 2007, and 2017, respectively.

### 4.2. Sediment Characteristics

The grain size of the seafloor sediment varied from the shoreline to the open water (Figure 5). The mean particle size (Mz) was 63.7 μm at sampling location 1 and 25.5 μm at sampling location 2, classifying the sediment as fine sand (63–125 μm) (Table 1). The grain size distribution frequency curves of samples #1 and #2 show multiple peaks and two peaks, respectively, whereas they both have the first peak at 0.59 μm (Figure 5).

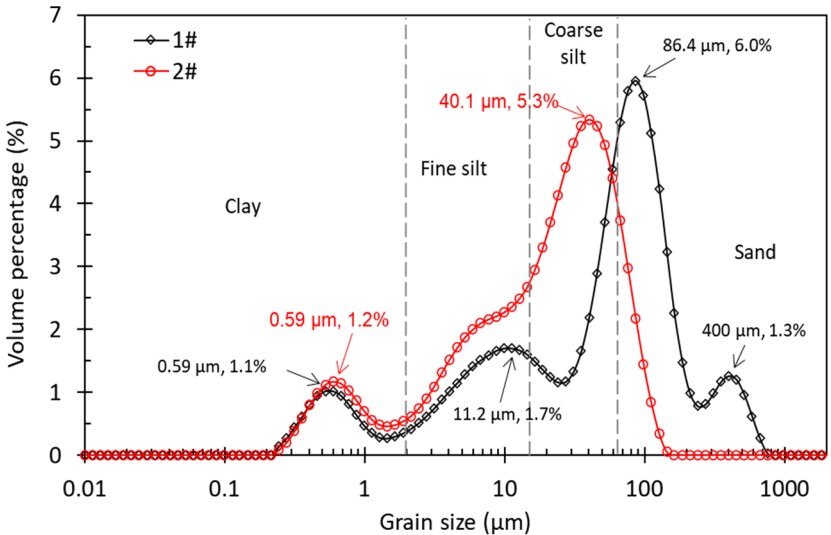

**Figure 5.** Grain size distribution curves of the seabed surface material.

**Table 1.** Grain size characteristics of the seabed surface material.

| Number | $D_{50}$ μm | Mz μm | 2 μm Clay | 2–16 μm Fine Silt | 16–63 μm Coarse Silt | 63–125 μm Fine Sand | 125–256 μm Medium Sand | >256 μm Coarse Sand |
|---|---|---|---|---|---|---|---|---|
| 1 | 58.9 | 63.7 | 9.67% | 19.47% | 22.75% | 27.87% | 12.97% | 7.3% |
| 2 | 21.2 | 25.5 | 11.57% | 28.04% | 48.79% | 11.19% | 0.39% | 0% |

Sediment load at the Lijin station showed that the Yellow River Delta received a total of $273 \times 10^8$ t of sediment through the SXG channel during 1953–1964 and through the DKH channel during 1964–1976, which contributed to forming the SXG-DKH lobes during the period (Figure 6).

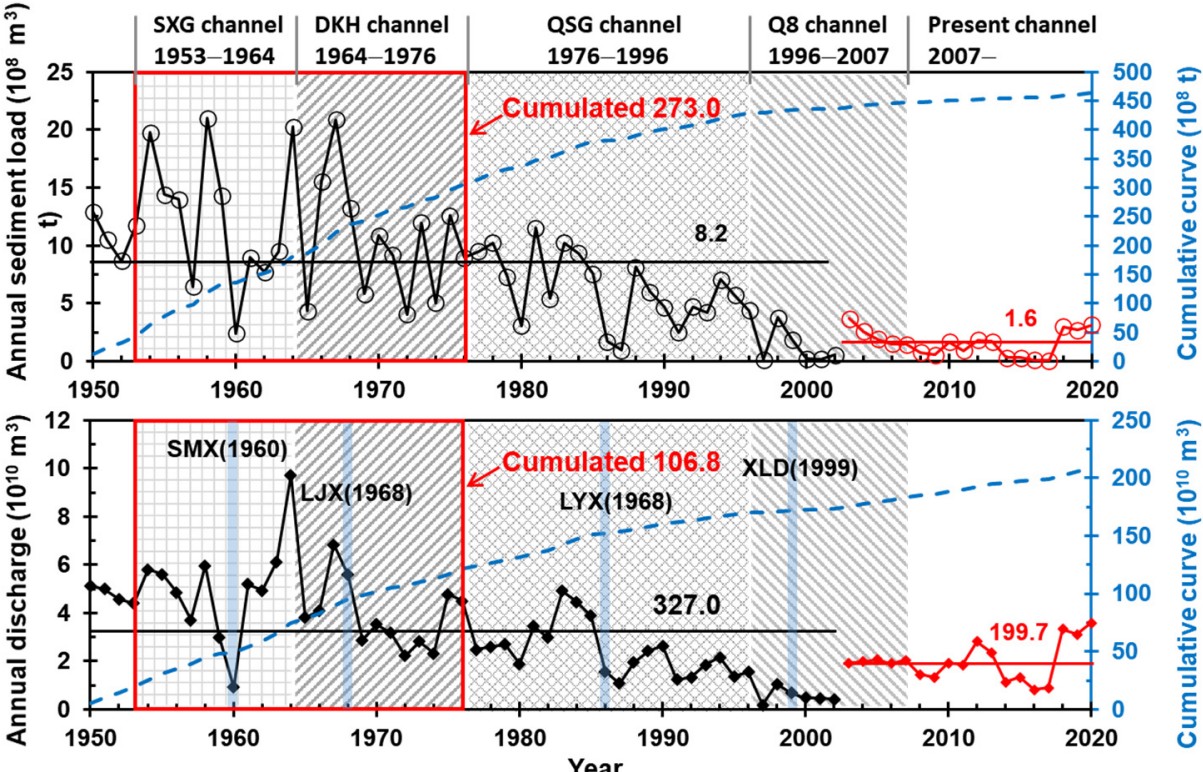

**Figure 6.** Sediment load and water discharge at the Lijin station. SMX is the Sanmenxia Reservoir, LJX is the Liujiaxia Reservoir, LYX is the Liuyangxia Reservoir, and XLD is the Xiaolangdi Reservoir. The red box represents the data from 1953 to 1976. The four gray wire frames represent the sediment and flow discharge data of the SXG (Shenxiangou) channel in 1953–1964, the DHK (Diaohekou) channel in 1964–1976, the QSG (Qingshuigou) channel in 1976–1996, the Q8 (Qing 8) channel in 1996–2007, and the present channel after 2007.

### 4.3. Seabed Surface Microtopography Features in September 2019

The multibeam data show that the microtopography of the seabed is uneven due to erosion (Figure 7a). The relative height difference between the lowest and highest points is about 1–2 m at a length of ~200 m (Figure 7a,e). These microtopographies have irregular edges and sudden depth change (Figure 7a,c–e). Three profiles across the areas showed that erosion phenomena were frequent (Figure 7b). The slope of these topographies could vary from 0 to 16° at a length of 70–200 m (Figure 7f–h).

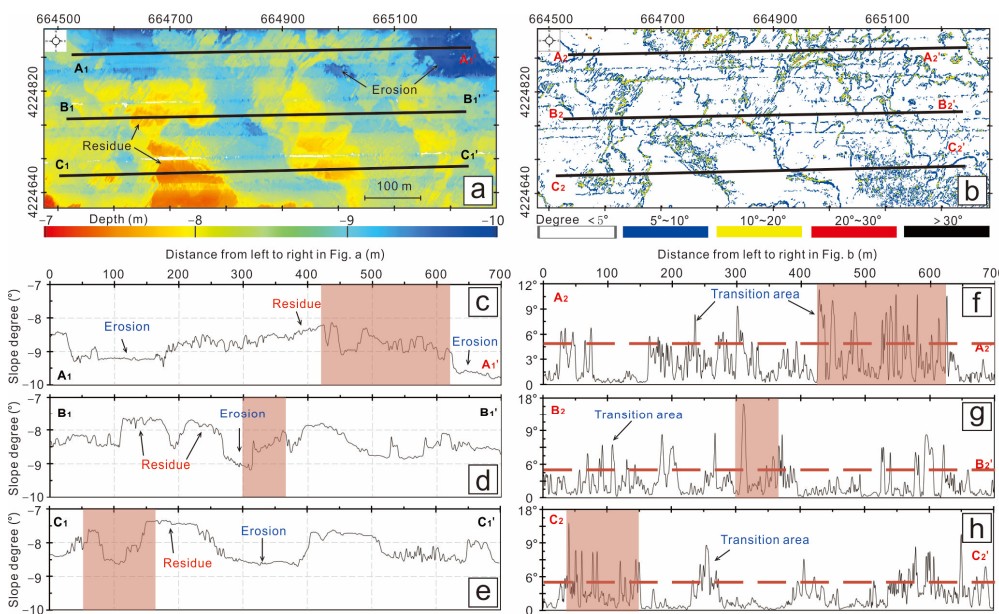

**Figure 7.** Maps showing the surface microtopography of the seabed (**a**) and its slope (**b**) with a resolution of 1 m × 1 m. A1–A1′, B1–B1′, and C1–C1′ in (**a**) show the microtopography profiles of the seabed in (**c–e**). A2–A2′, B2–B2′, and C2–C2′ in (**b**) show the slope profiles of the seabed in (**f–h**). The red dotted lines are slope degree of 5°.

## 5. Discussion

The deformation of the nearshore seabed can also be influenced by changes in the sediment supply [42–44], tidal cycles and sea level rise [45], land subsidence [44,46] and anthropogenic disturbances [46–48], and river course migration [6]. In addition, high-energy storm events are also one of the most important factors influencing topographic changes in the seabed near coastlines [49]. Therefore, similar to other large river deltas in the world, the evolution mechanism of the Yellow River Delta is complicated [47], especially for the abandoned subdelta [28,48].

Our study suggests that overall, the seabed of the study area has a net volume of $5.3 \times 10^8$ m$^3$ (with area of $3.6 \times 10^8$ m$^2$) of sediment which was eroded from 1984 to 2017. The erosion of the seabed may be related to the interruption of the sediment supply caused by the migration of the river course, as well as sea level rise and land subsidence and anthropogenic disturbances [44]. Similar situations were found in the Lower Yellow River and the delta and support this hypothesis [28,42–44]. For example, a channel incision was observed in the Lower Yellow River owing to upstream damming [42], and the reduction in riverine sediment load caused the transition of the Yellow River Delta from a depositional phase in 1980–1998 to an erosional phase after 1998 [43]. Human projects such as embankments along the coastal area may have prevented coastline retreat, leading to severe subaqueous seabed scouring. Fan et al. (2018) reported that the coastline retreated rapidly and that the corresponding subaqueous area decreased rapidly near the Dongying port (Figure 1c) [28]. However, it is worth noting that the subaqueous area near Dongying port changed only moderately (from 360 km$^2$ to 370 km$^2$) due to the construction of the artificial coastline (Figures 1c and 3c). This value was much less than the change in the northern area of the SXG lobe (more than 11 km during 1976–2020 [49]). Nevertheless, the subaqueous part of the SXG-DKH lobes showed that the area from 0 to −5 m decreased during 1984–2017; simultaneously, the areas from −5 to −10 m and below −10 m increased (Table 2 and Figure 3c). However, the void volume from the water surface to the seafloor showed an increasing trend, with annual average increases of $9.33 \times 10^6$ m$^3$ from 1984–2007 and $3.95 \times 10^6$ m$^3$ from 2007–2017 (Table 2). A similar increasing trend was also found in the areas within −5−−10 m and below −10 m (Table 2). The water depth changes from 2007–2017 also support the void volume from the water surface to the seafloor increase in

the study area (Table 2). Therefore, although human projects fixed and prevented coastline erosion, the subaqueous topography experienced severe scouring.

**Table 2.** Changes in the subaqueous seabed of the SXG-DKH lobes during 1984–2017.

| Parameters | Year | From 0 to −5 m | From −5 to −10 m | Below −10 m |
|---|---|---|---|---|
| Annual volume change | 1984–2007 | 9.33 | 5.87 | −0.7 |
| ($\times 10^6$ m$^3$) | 2007–2017 | 3.95 | 11.7 | 4.2 |
| Annual depth change | 1984–2007 | 4.38 | −1.15 | −1.97 |
| (cm) | 2007–2017 | 4.21 | 3.15 | −0.14 |
| Annual area change | 1984–2007 | −1.31 | 0.95 | 0.92 |
| ($\times 10^6$ m$^2$) | 2007–2017 | −2.61 | 0.77 | 1.34 |

In addition, seabed erosion is generally accompanied by sediment coarsening [34,42]. Thus, the surface seabed sediment coarsening trend can prove the erosion of the seabed. Although the sediment samples from this study cannot support the sediment coarsening due to there being only two samples collected (Table 1), the seabed sediment of the Yellow River Delta coast has coarsened since the 1990s [50]. As an example, the average mean grain size of the abandoned delta lobe increased from 28.6 μm in 1992 to 35.7 μm in 2000 [51]. Similar seabed erosion and seabed sediment coarsening patterns can also be found in the lower Yellow River [42]. Moreover, the multiple peaks of sediment samples indicate strong seabed sediment resuspension and transport [34]. In this study, we found that the two surface seabed sediments have multiple peaks (Figure 5). This phenomenon is related to strong wave action caused by protuberant topography [34]. The above studies and our finding of the grain-size characteristics of surface seabed sediment support the strong seabed sediment transport. Besides, the average rate of sea level rise will increase seabed deformation [14,15]. In this study, our results regarding seabed deformation support this point (Table 2 and Figure 3). Hence, the erosion trend of the subaqueous SXG-DKH lobes is expected to continue due to the interruption of the sediment supply, human projects, and sea level rise. It is not clear if seabed erosion of the abandoned delta lobes would have any effect on the stability of the coastal shoreline and continental shelf.

## 6. Conclusions

This study assessed 33-year seabed deformation in an abandoned subdelta on the Yellow River Delta of the Bohai Sea, China. We found that the void space from the sea water surface to the seafloor within an area of $3.6 \times 10^8$ m$^2$ increased by 16% from 1984 to 2017, totaling a net erosion of $5.3 \times 10^8$ m$^3$ of seabed materials. This volumetric quantity is equivalent to 6.89 billion metric tons, assuming a bulk density of 1.3 t/m$^3$ for the seabed sediment. The erosion from 0 to −5 m, from −5 to −10 m, and below −10 m has showed a similar increasing trend over the past 33 years. The subaqueous topography experienced severe scouring. Erosional microtopography was found in the SXG-DKH seabed, with a relative maximum depth difference of ~1 m. These findings suggest that seabed erosion in the abandoned subdelta in the Yellow River Delta will likely continue, and that other abandoned delta lobes in the world may also have been experiencing similar seabed erosion due to interrupted sediment supply and sea level rise. Further studies are needed to discern whether seabed erosion of abandoned delta lobes in the world would have any impact on the stability of the coastal shoreline and continental shelf.

**Author Contributions:** Writing—original draft preparation, S.Z. and Y.Z. (Yunfeng Zhang); writing—review and editing, Y.X. and Y.Z. (Yuyan Zhou); data curation, Y.C.; methodology, C.H. and H.C.; software, C.H.; formal analysis, S.G.; investigation, S.Z. and Z.L., funding acquisition, S.Z. and Y.Z. (Yuyan Zhou). All authors have read and agreed to the published version of the manuscript.

**Funding:** This research was funded by the Open Found Project of the Shandong Provincial Bureau of Geology and Mineral Resources, grant number 2019KF801-5, the Natural Science Foundation of Shandong Province, grant number ZR2020QD083 and 51909275, the Qinghai Central Government Guided Local Science and Technology Development Fund Project, grant number 2022ZY020, the IWHR Research & Development Support Program, grant number WR110145B0052021, and the Open Research Fund of State Key Laboratory of Simulation and Regulation of Water Cycle in River Basin, IWHR, grant number IWHR-SKL-KF202204. During the preparation of this manuscript, Y.J.X. received a U.S. Department of Agriculture Hatch Fund project (Project#: LAB94459).

**Data Availability Statement:** Not applicable.

**Acknowledgments:** The authors are grateful to the associate editor and two anonymous reviewers for their valuable feedback and suggestions, which were important and helpful to improve the quality of this manuscript.

**Conflicts of Interest:** The authors declare no conflict of interest.

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
