# Peer review of "Subaqueous Topographic Deformation in Abandoned Delta Lobes—A Case Study in the Yellow River Delta, China"

_water, doi:10.3390/w15112050_

Round 1

Reviewer 1 Report

Comments can be found in the appended file.

Author Response

Revision Notes (point by point)

Reviewer #1

Overall assessment

This article studies the bottom structure, sediment and dynamics of an abandoned subdelta of the Yellow River Delta. It is overall reasonably well written (in terms of language), but not consistent enough, not well structured (some results are only presented in the Discussion section), and badly referenced (the references are a mess and it is impossible to check if all of them are appropriate, relevant and supportive of the paper’s statements and conclusions.

[Authors’ response] Thanks for taking your valuable time to review our manuscript and provide helpful comments and suggestion, all of have been taken into account while revising our manuscript. As suggested, we have done some restructuring to make the results and our discussion points clearer. We have done our best to improve accuracy of citations to discuss and support the findings from our study. Your comments and suggestions have been helpful for us to improve the quality of our manuscript, for which we are grateful.

Some of the conclusions seem speculative, considering the data available, and the overall impression of a 30-year data series of bathymetric data given in the abstract is misleading, as only 3 bathymetric surveys (plus a partial microtopography survey) were used. The article uses bathymetries from 1984 and 2007 and 2017, and a high-resolution bathymetry obtained in 2019 for a part of the study area, to make inferences about the bottom dynamics for a 30-year period. But, in my opinion, data do not support the inferences about a “continuous trend” (as we do not know what happened between surveys) and the “strong suggestion that seabed erosion will continue” (suspended sediment data may

allow that inference, but 3 bathymetry maps not).

[Authors’ response] Your point is well taken. We have modified the Results, the Discussion, the Conclusions and the Abstract to meet your advice. Please see the revised version.

The methods are not clearly described, nor are their limitations discussed. For instance, it is not clear what the kriging is used for. Is it to interpolate the MBES data? The microtopography map clearly shows the survey tracks, suggesting errors in the bathymetry corrections (i.e. the roll, pitch and yaw correction and the tide correction). These errors need to be discussed.

[Authors’ response] Your point is well taken. We have rewritten the methods section to provide more information. The following text has been added in the revised manuscript (please see lines 152-169).

“In this study, the bathymetric charts were georeferenced using 6-9 fixed landmarks in ArcGIS 10.3 [4]. Subsequently the water depth points were collected from bathymetric charts. These points including original water depth points (adjacent cross-sections were spaced 1 km apart, having 250-450 m intervals along each cross-section) and water depth points on the isobaths of 0 m, -2 m, -5 m, -10 m, and -20 m (each point-intervals along isobaths were approximately 300-500 m). As a result, each digitized bathymetric chart has a density of 70-100 points per square kilometer. The ordinary kriging interpolation method was used to generate a digital elevation model (DEM) with a resolution of 100 × 100 m in ArcGIS 10.3 (ESRI, Redlands, CA). The Kriging method has been widely used and is highly effective in the analysis of volume from water surface to the seafloor [35-37]. The interpolation error is mainly from bathymetric chart data and the interpolation method (kriging interpolation method). Overall, the error of the DEMs was estimated to be less than 10% [9]. In addition, the mean rate of sea level rise (this value including sediment compaction and land subsidence) is 3.6 mm/a during 1980-2021 in the study area (https://www.mnr.gov.cn). In this study, we select this value as the seabed volume change caused by sea level rise and land subsidence. The study aera is of 3.6× 108 m2. Thus, the seabed volume change caused by sea level rise and land subsidence (Vs) is 0.31 × 108 m3 and 0.14 × 108 m3 during 1984-2007 and 2007-2017, respectively.”

The data available could be better exploited. The 2019 micro-bathymetry survey could be compared with the 2017 bathymetry to assess the most recent dynamics within the 500x800m rectangle surveyed in 2019. Also, the sediment size analysis results are hardly explored. I wonder if they contribute enough to the article to deserve being included.

Being very descriptive, not innovative, badly referenced and with lack of some important technical information, I consider that the article should not be published in its current form. Maybe after some major review. Detailed comments are provided below (L = manuscript line).

[Authors’ response] Thank you for the suggestions. In the revised manuscript, We have rewritten the Discussion part to topography, sediment size and checked the references one by one. Please see the revised version.

References

The references should be thoroughly reviewed. There are different references with the same number (which seem to be later-inserted references). This is not acceptable. Renumber references according to their appearance in the text and without duplicating reference numbers. Furthermore, there are many references in the text that cannot be found in the list.

[Authors’ response] Thank you for catching the errors. In the revised manuscript, we have added, checked and modified the references one by one.

L46 Please check your references. For instance, I can’t find the information cited here in reference [18].

[Authors’ response] Sorry for the mistake. The correct reference is “Nienhuis, et al., 2020”. In the revised manuscript, we have carefully checked all references and renumbered them accordingly..

Nienhuis, J.H., Ashton, A.D., Edmonds, D.A., Hoitink, A.J.F., Kettner, A.J., Rowland,J.C., T?rnqvist, T.E. Global-scale human impact on delta morphology has led to net land area gain. Nature. 2020, 577(7791): 514-518.

L120/121 “The vertical error in the depth is 0.2 m at water depth is ≤10 m and ±0.1 H (H= water depth) at water depth >20 m [9].” Apart from the poor English of this sentence, this information cannot be found in reference [9]. This information is cited in reference [34].

[Authors’ response] Thank you for catching this. We have modified the reference and have rephrased the sentence to: “The vertical error of water depth is 0.2 m and ±0.1 H, when H≤10 m and H>20 m, respectively [9].”

Now, the reference [9] changed to: Zheng, S., Cheng, H., Shi, S., Xu, W., Zhou, Q., Jiang, Y., Zhou, F., Cao, M. Impact of anthropogenic drivers on subaqueous topographical change in the Datong to Xuliujing reach of the Yangtze River. Sci. China Earth Sci, 2018, 61: 940-950.

L147-150 You write “In this study, the ordinary kriging interpolation method was used to generate a digital elevation model (DEM) with a resolution of 100 × 100 m by using ArcGIS 10.3 (ESRI, Redlands, CA). The method has been widely used and is highly effective in the analysis of void volume from water surface to the seafloor [34-36]“. However, references [35] and [36] do neither use kriging nor ArcGIS.

[Authors’ response] Sorry for the incorrect referencing. We have corrected the reference numbers. We have checked the Method section of these references and have made sure that they have used the Kriging interpolation method. The corrected references of [34-36] in the revised manuscript are as follows:

  1. Zheng, S., Cheng, H., Lv, J., Li, Z., Zhou, L. Morphological evolution of estuarine channels influenced by multiple anthro-pogenic stresses: A case study of the North Channel, Yangtze estuary, China. Estuar. Coast. Shelf S. 2021. 249, 107075.
  2. Dai, Z., Liu, J. T., Fu, G., & Xie, H. A thirteen-year record of bathymetric changes in the North Passage, Changjiang (Yangtze) estuary. Geomorphology. 2013, 187: 101-107.
  3. Xu, Y.J., Wang, B., Xu, W., Tang, M., Tsai, F.T.C., Smith, L.C. Four-decades of bed elevation changes in the heavily regulated upper Atchafalaya River, Louisiana, USA. Geomorphology. 2021, 386: 107748.

L224/225 I think this statement is not supported by reference [38].

[Authors’ response] You are correct. We have removed this sentence and reference. Now this part has been rewritten to: “The multibeam data showed that the seabed is unevenness with erosion and residue microtopography (Fig. 5a). The relative height difference between the lowest and highest points is about 1−2 meters on a length of ~200 m (Fig. 5a and 5e). These micro topographies have irregular edges and sudden depth change (Fig.5a, 5c-e). Three pro-files across the areas showed that erosion phenomena were frequent (Fig. 5b). The slope of these topographies can changed from 0−16°on a length of 70−200 m (Fig. 5f-h).” Please see lines 261-266.

L287/288 There are many references missing, e.g. references [46-51].

[Authors’ response] We apologize for the mistakes. We have now correctly numbered the references.

Language

L 19 “…which has 3-decade long (1984-2017) bathymetric data and long-term river discharge and sediment load record” should be changed to “… for which 3-decade long (1984-2017) bathymetric data and long-term river discharge and sediment load records are available.”

[Authors’ response] Thank you for your kind help. We have revised this sentence as you suggested. Please see lines 19-21.

L23 “We found that the study seabed increased a net volume of 5.8 × 108 m3 (with area of 3.6× 108 m2) from 1984 to 2017.” The seabed lost sediment and, hence, decreased in volume. The water mass above increased in volume. The way the results are presented may be confusing. You could consider presenting sediment budget instead of above-bottom water volume change.

[Authors’ response] Thank you for the constructive suggestion. We have revised the sentence to: “We found that a net volume of 5.8 × 108 m3 (with area of 3.6× 108 m2) of sediment was eroded from the study seabed during 1984−2017.” Please see line 24-25.

L28 “The subaerial topography experienced severe scouring.” Redundant as it basically means that there was severe subaerial (seabed) erosion, as mentioned before.

[Authors’ response] Deleted.

L42 Due to “sea level”, or “sea level rise”/”sea level changes”?

[Authors’ response] Thank you; the suggested change has been made.

L60 Consider writing “…to gain knowledge about long-term …”

[Authors’ response] Done, thank you for the suggestion.

L69/70 Do you mean “… strategies and plans …”_

[Authors’ response] Yes, corrected.

L83/84 “For example, the most of abandoned Yellow River deltas had turned to slight erosion after 2000 [1].” Should read “For example, most of the abandoned Yellow River deltas have turned to slight erosion after 2000 [1].”

[Authors’ response] Thank you for your kind help. Changed.

L84/85 Consider writing “On the whole, the subaqueous Yellow River Delta experienced a change from deposition to erosion after 2005 [22].”

[Authors’ response] Thank you for your kind help. Modified.

L102/103 Do you mean the mean tidal range and mean significant wave height?

[Authors’ response] Yes, revised.

Figure 1. Caption – “The colored lines show the lowermost channel shifts.” Do you mean “The colored lines show the most recent channel shifts”?

[Authors’ response] Thank you for catching this. The suggested change has been made.

Figure 1. Caption – I think you can remove the “MBES: multibeam echo sounder” from the legend and the figure. It looks in the figure as if this equipment was placed somewhere in the sea. It is enough to mention that “The white rectangle with a black border shows the MBES (multibeam echo sounder) survey area in September 2019.”

[Authors’ response] We agree. Removed.

L147-150 Please indicate what data were interpolated using kriging and what type of kriging was applied (ordinary, simple, universal, …) and what model.

[Authors’ response] Thank you for your helpful suggestion. We have rewritten this part to: “In this study, the bathymetric charts were georeferenced using 6-9 fixed landmarks in ArcGIS 10.3 [4]. Subsequently the water depth points were collected from bathymetric charts. These points including original water depth points (adjacent cross-sections were spaced 1 km apart, having 250-450 m intervals along each cross-section) and water depth points on the isobaths of 0 m, -2 m, -5 m, -10 m, and -20 m (each point-intervals along isobaths were approximately 300-500 m). As a result, each digitized bathymetric chart has a density of 70-100 points per square kilometer. The ordinary kriging interpolation method was used to generate a digital elevation model (DEM) with a resolution of 100 × 100 m in ArcGIS 10.3 (ESRI, Redlands, CA).” Please see lines 152-160.

L179/180 “Eight MBES survey lines perpendicular to the isobath direction were conducted near the DKH subdelta lobe (1500 m × 3500 m) on 28 September 2019.” In lines 127/128 you mention that the surveyed area is 500 by 800m.

[Authors’ response] Sorry for the confusion. The MBES survey area is 500 m ×800 m, while the area of studied DKH subdelta lobe is 1500 m×3500 m. To avoid confusion, we have revised the sentence to: “Eight MBES survey lines perpendicular to the isobath direction were conducted near the DKH subdelta lobe on 28 September 2019.” Please see lines 193-194.

Figure 4 and Table 1 Should use “µm” instead of “um” for micrometre

[Authors’ response] We use “µm” as the grain size is given in the unit.

Figure 7 “green blocks”?

[Authors’ response] Thank you for catching this. Revised.

Content

L137 When was the sediment sampled, in 2019?

[Authors’ response] Yes, it was sampled in 2019. In the revised manuscript, we have added this information.

L180/181 “The roll, pitch and yaw were calibrated, and the error beam was removed …” What is an error beam? Do you mean that measurement errors due to roll, pitch and yaw were corrected? So you used an IMU? This should be mentioned.

[Authors’ response] Sorry for the confusion. We have rewritten this sentence to: “The water depth data collected by MBES were processed in the PDS 2000 software (PDS V4.1.7.3), including beam calibrations for roll, pitch and yaw, as well as error beam remove.” Please see lines 194-196.

Figure 5 – The bathymetry in this figure does nor seem to correspond to the bathymetry in the white MBES rectangle of Figure 1. Figure 1 and L179/180 suggest that the profiles are drawn perpendicular to the coast and isobaths. I would, therefore, expect another bathymetry map in Figure5a, with depths increasing from left to right.

[Authors’ response] Sorry for the confusion. Figure 5 does not show the orientation of “up-north and down-south”, please refer to the direction of its north arrow. We have revised the “north arrow” and made it more clear in the figure.

Discussion – I think Figure 6 and, particularly Figure 7 do not belong in the discussion. These seem to be results.

[Authors’ response] Thank you for your kind suggestion. We have moved Figure 6 and Figure 7 into the Results section.

Discussion – References [46-51] are missing in the list. It is therefore impossible to verify many of the author’s statements.

[Authors’ response] Thank you for catching this. The references have been carefully renumbered.

L238/239 “Our study suggests that the seabed of the SXG-DKH lobes continuously eroded from 1984 to 2017.” The decreased sediment load may suggest that, but a study based on 3 observations (form 1984, 2007 and 2017) cannot suggest a continuous erosion along 30 years. Especially if you later say (L240-242) “[42] insisted that the reduction in riverine sediment load caused the transition of the Yellow River Delta from a depositional phase in 1980-1998 to an erosional phase after 1998.”, suggesting that the erosional trend started in 1998 and not in 1984.

[Authors’ response] We agree with you. We have rewritten this sentence to: “Our study suggests that overall, the seabed of study area has a net volume of 5.3 × 108 m3 (with area of 3.6× 108 m2) of sediment was eroded from 1984 to 2017.”

L272-274 “Three cross-section profiles (P1-P3, locations in Fig.1 c) showed that the -5 m and -10 m isobaths seem to prograde toward the coastline, and this trend increased suddenly after 2007 (Fig. 7).” The data do not allow to state that this increase occurred suddenly after 2007. It may have occurred gradually or suddenly, sometime between 2007 and 2017.

[Authors’ response] Thank you for pointing this out. We have deleted the string “, and this trend increased suddenly after 2007”.

L285/290 “Seabed erosion is generally accompanied by sediment coarsening [41,50]. The seabed sediment of the Yellow River Delta coast has coarsened since the 1990s, and this coarsening trend has intensified in recent years [51]. As an example, the average mean grain size of the abandoned delta lobe increased from 28.6 µm in 1992 to 35.7 µm in 2000 [51]. Our data on the seabed sediment from the SXG-DKH lobes also support this conclusion (Table 1).” I can’t confirm that, since there are no references [50] and [51] in the reference list. The data of this study show mean grain sizes (MZ) of 63.7 mm and 25.5 mm for sample points #1 and #2, respectively, in 2019 (I suppose it was in 2019, as it is not mentioned). Depending on where the sediment samples of reference [51] were collected, this may or may not corroborate the conclusion.

[Authors’ response] We agree. First, we have renumbered the references and carefully checked the references. The data from reference [51] (in the revised manuscript is [52]) indicates that the average mean grain size of the abandoned delta lobe increased from 28.6 µm in 1992 to 35.7 µm in 2000. Second, we have deleted the sentence: “Our data on the seabed sediment from the SXG-DKH lobes also support this conclusion (Table 1)”.

Thank you again for your kind help and suggestions. We have done our best to improve the manuscript and hope that the changes made meet your expectations.

Reviewer 2 Report

Dear authors  

I have read the manuscript entitled “Subaqueous topographic deformation in abandoned delta 2

lobes – A case study in the Yellow River Delta, China” by Zhang et al.

The manuscript deals with an issue (evolution of an abandoned lob within the Yellow river delta) suitable for an international audience. Nevertheless, is not publishable in the present form and needs of major revisions.

The authors analyze a data set consisting of three decades long bathymetric data, a long-term river discharge and sediment load record. They implemented this data set by means of a new bathymetric multibeam survey evidencing a  severe erosion of the seabed surface. 

 In my opinion, there are several weaknesses (also conceptual) in the manuscript. 

·      The authors analyze a data set consisting of three decades long bathymetric data, a long-term river discharge and sediment load record. They implemented this data set by means of a new bathymetric multibeam survey evidenced severe erosion of the seabed surface. 

·      “Subaerial seabed” is an oxymoron.  

·      The meteo-marine (tides, prevalent winds, waves) conditions of the study area (and as a consequence the direction of the main sedimentary drift) are not reported. This information is of fundamental importance to understand the modality of the morphological changes (“Deformation”).

·      In the quantification of seabed “deformation” (chapter 3.3) it is not considered the rate of the sediment compaction. This value like you made about the rate of relative sea level rise (summation between the eustatic rise and subsidence) should be quantified. Is it within the error bar of the DEMs?  And in the same way is the value rate of relative sea level rise (summation between the eustatic rise and subsidence) within the error bar of the DEMs?

·      Regarding the grain size characteristic of seabed “surface material” (are sediment not material) it seems that only two samples have been collected. Normally, for a grain size characterization, many samples should be collected along transects more or less parallel to the main direction of the sedimentary drift. So, the data presented are not enough for general consideration.

·       In the 4.3 chapter the new data are not analyzed in depth as they deserve.  In morphodynamical terms what is the meaning of the edges? Are the sudden changes of depth due to channelization processes? What are the main directions of the sedimentary drift?

·       Your results are however expected. From a general point of view, abandoned lobe is almost always characterized by erosion processes because of the deactivation of mouth channel and the consequent strong diminution of local sediment supply. This is irrespective of the entire Delta system sediment supply.  

Other key points to improve regards the figures.

·       A satellite or google earth imagine of the Yellow River delta is needed. Can be added in Fig. 1 for example.

·       Fig. 1. Insert the scale in Fig. 1 a

·       Fig. 4 Insert in the caption the meaning of the red and black  lines.

References

·       There are some mistakes (In between 36 and 37 there is a caption numbered  as 5 The same in between 44 and 46--  6) Please check .

Author Response

Revision Notes (point by point)

Reviewer #2

Dear authors:

I have read the manuscript entitled “Subaqueous topographic deformation in abandoned delta lobes – A case study in the Yellow River Delta, China” by Zhang et al.

The manuscript deals with an issue (evolution of an abandoned lob within the Yellow river delta) suitable for an international audience. Nevertheless, is not publishable in the present form and needs of major revisions.

[Authors’ response] First of all, we thank you very much for taking your valuable time to review our manuscript. Your comments and suggestions are greatly appreciated.

The authors analyze a data set consisting of three decades long bathymetric data, a long-term river discharge and sediment load record. They implemented this data set by means of a new bathymetric multibeam survey evidencing a severe erosion of the seabed surface.

In my opinion, there are several weaknesses (also conceptual) in the manuscript.

The authors analyze a data set consisting of three decades long bathymetric data, a long-term river discharge and sediment load record. They implemented this data set by means of a new bathymetric multibeam survey evidenced severe erosion of the seabed surface.

[Authors’ response] Thank you for your careful reading. In the revised manuscript, we have done our best to address your concerns, based on the findings from our data as well as those from published results. To incorporate your points, the Title, Abstract, Introduction, and Discussion section have been almost completely rewritten. We believe the changes/additions made are a substantial improvement and hope that they meet your expectations.

“Subaerial seabed” is an oxymoron.  

[Authors’ response] Thank you for catching this. We have removed “subaerial” before “seabed”throughout the manuscript.

The meteo-marine (tides, prevalent winds, waves) conditions of the study area (and as a consequence the direction of the main sedimentary drift) are not reported. This information is of fundamental importance to understand the modality of the morphological changes (“Deformation”).

[Authors’ response] Thank you for your comment. As suggested, we have added the following text in the revised paper. 

“It belongs to warm temperate monsoon climate zone, with annual mean rainfall of 590.9 mm [16]. The southerly winds prevail with a range of 3-6 m/s in summer and northerly winds prevail with >10 m/s in winter [30]. The mean wave height up to 1.5 m [30]. The mean tidal near the shoreline range of 0.6-0.8 m in the YRD [16].” Please see lines 97-101.

In the quantification of seabed “deformation” (chapter 3.3) it is not considered the rate of the sediment compaction. This value like you made about the rate of relative sea level rise (summation between the eustatic rise and subsidence) should be quantified. Is it within the error bar of the DEMs?  And in the same way is the value rate of relative sea level rise (summation between the eustatic rise and subsidence) within the error bar of the DEMs?

[Authors’ response] This is a good point and thank you for bring it up. In the revised manuscript, we have added the influence of sea level rise and land subsidence in the volume change calculation of the seabed deformation. Accordingly, we have updated the results throughout the manuscript to reflect their effects. Please see Abstract, Results, Discussion, and Conclusions part.

Regarding the grain size characteristic of seabed “surface material” (are sediment not material) it seems that only two samples have been collected. Normally, for a grain size characterization, many samples should be collected along transects more or less parallel to the main direction of the sedimentary drift. So, the data presented are not enough for general consideration.

[Authors’ response] Your point is well taken. We admit that it would be great if more sediment samples were taken. However, due to funding limitation, we only sampled at two locations. On the other hand, the grain size measurement does not affect the main findings from this study. Thank you for your careful reading.   

In the 4.3 chapter the new data are not analyzed in depth as they deserve.  In morphodynamical terms what is the meaning of the edges? Are the sudden changes of depth due to channelization processes? What are the main directions of the sedimentary drift?

[Authors’ response] Thank you for your comments. We have rewritten this part to:

“The multibeam data showed that the seabed is unevenness with erosion and residue microtopography (Fig. 7a). The relative height difference between the lowest and highest points is about 1−2 meters on a length of ~200 m (Fig. 7a and 7e). These micro topographies have irregular edges and sudden depth change (Fig.7a, 7c-e). Three profiles across the areas showed that erosion phenomena were frequent (Fig. 7b). The slope of these topographies can changed from 0−16° on a length of 70−200 m (Fig. 7f-h).” We hope the changes made meet your expectations. Please see lines 261-266.

Your results are however expected. From a general point of view, abandoned lobe is almost always characterized by erosion processes because of the deactivation of mouth channel and the consequent strong diminution of local sediment supply. This is irrespective of the entire Delta system sediment supply.  

[Authors’ response] Thank you for your comments. While the findings from our study reflect general point of view, they add more observations to the knowledge and specifically for the Yellow River Delta, which is often viewed as a prograding system.    

  • Other key points to improve regards the figures.

A satellite or google earth imagine of the Yellow River delta is needed. Can be added in Fig. 1 for example.

[Authors’ response] Thank you for the good suggestion. In the revised manuscript we have added a google map of the area.

Fig. 1. Insert the scale in Fig. 1 a

[Authors’ response] Added, as suggested.

Fig. 4 Insert in the caption the meaning of the red and black  lines.

References

[Authors’ response] Thank you; the suggested change has been made.

There are some mistakes (In between 36 and 37 there is a caption numbered  as 5 The same in between 44 and 46--  6) Please check .

[Authors’ response] Thank you for catching the mistakes. In the revised manuscript, all references have been carefully checked and renumbered.

Round 2

Reviewer 1 Report

The manuscript has been clearly improved, particularly in terms of results and discussion, and the missing information was added. However, to be recommended for publication the language needs to be checked, particularly in the newly inserted text. Just to provide some examples, in L262/263 the sentence “The multibeam data showed that the seabed is unevenness with erosion and residue microtopography (Fig. 7a).” has several mistakes, and in L266/267 the authors probably mean either “The slope of these topographies can change…” or “The slope of these topographies could vary from 0 to 16° on a length of 70−200 m (Fig. 7f-h)." Also I believe the minus sign is not “-” but “-".
